# The Effect of Temperatures and Hosts on the Life Cycle of *Spodoptera frugiperda* (Lepidoptera: Noctuidae)

**DOI:** 10.3390/insects13020211

**Published:** 2022-02-20

**Authors:** Yi-Chai Chen, De-Fei Chen, Mao-Fa Yang, Jian-Feng Liu

**Affiliations:** 1State Key Laboratory Breeding Base of Green Pesticide and Agricultural Bioengineering, Key Laboratory of Green Pesticide and Agricultural Bioengineering, Ministry of Education, Guizhou University, Guiyang 550025, China; yichaichen@126.com; 2Guizhou Provincial Key Laboratory for Agricultural Pest Management of the Mountainous Region, Scientific Observing and Experimental Station of Crop Pest in Guiyang, Ministry of Agriculture, Institute of Entomology, Guizhou University, Guiyang 550025, China; gdgdly@126.com; 3Crop Protection Center of Jinsha County, Bijie 551700, China; chendefei2012@163.com; 4College of Tobacco Science, Guizhou University, Guiyang 550025, China

**Keywords:** *Spodoptera frugiperda*, host plants, temperature, age-stage, two-sex life table, demographic parameters

## Abstract

**Simple Summary:**

Fall armyworm (FAW), *Spodoptera frugiperda* (J E Smith) (Lepidoptera: Noctuidae) is an invasive worldwide agricultural pest that seriously threatens the safety of grain production. Temperatures and hosts play vital roles in the performance of *S. frugiperda*. In this study, we studied the effects of three different temperatures (20, 25, and 30 °C) and three different host plants (maize (*Zea mays* L. (Poaceae)), sorghum (*Sorghum bicolor* (L.) Moench), and coix seed (*Coix lacryma-jobi* L.)) on the whole life cycle of *S. frugiperda*. The results showed that both temperatures and host plants significantly influenced the mortality, developmental time, reproduction, and population parameters of FAW. In the treatment with host plants, the developmental period of *S. frugiperda* at each stage shortened significantly with the increasing of temperatures. At each temperature, the type of host plant did not affect the development time of *S. frugiperda* eggs. However, there was no significant difference when *S. frugiperda* 1st to 6th instar larvae were fed different host plants at 20 °C, while 2nd to 5th instar larvae developed rapidly on maize at 25 and 30 °C. Moreover, there were no significant differences in the longevity of FAW when fed sorghum and coix seed at 20 and 25 °C, but a significant difference in the longevity was observed at 30 °C. Feeding on maize achieved a higher intrinsic rate of increase (*r*), finite rate of increase (*λ*), and net reproductive rate (*R*_0_) of *S. frugiperda* at 25 and 30 °C than those FAW fed on sorghum and coix seed.

**Abstract:**

The interactions between ambient temperatures and host plants are central to the population dynamics of invasive animal species. Despite significant research into the effects of temperatures, the performance of invasive species is also influenced by host plants. The effects of different temperatures (20, 25, and 30 °C) and host plants (maize, sorghum, and coix seed) were tested on the mortality, development, reproduction, and population parameters of the fall armyworm (FAW), *Spodoptera frugiperda* (J E Smith) (Lepidoptera: Noctuidae), using an age-stage, two-sex life table. The results support the hypothesis that temperature and the species of the host plant significantly influences the performance of FAW. Feeding on maize at 30 °C resulted in a lower mortality rate, a shorter developmental time and longevity, a higher fecundity, intrinsic rate of natural increase (*r*), finite rate of increase (*λ*), and net reproductive rate (*R*_0_). However, at 20 °C, the host plant could eliminate temperature-mediated synergism in FAW performance, which did not reach statistical significance at 20 °C. Similar results induced by a relatively low temperature (20 °C) on different host plants were also found in the age-stage specific survival curves (*s_xj_*), fecundity (*m_x_*), maternity (*l_x_m_x_*), and reproductive value (*v_xj_*) curves of FAW. Consequently, we also need to pay more attention to FAW outbreaks on different host plants mediated by relatively low temperatures.

## 1. Introduction

Global climate change may substantially affect the diverse geographical distributions of organisms, alter biodiversity, and influence the interactions between species and ecosystems [1]. Invasive species can be more capable than native species at responding to climate change due to their high fecundity and abundance [2,3]. Temperature-driven changes are thought to alter ecosystem stability and sustain generational cycles for multivoltine insects as temperatures increase [4]. Therefore, increasing temperatures may expand the overall global distribution of invasive crop pest species, triggering outbreaks, and, in turn, increasing the damage to crops [4,5,6]. The establishment and spread of invasive pest species in new environments are commonly attributed to a lack of natural enemies. However, the lack of shared evolutionary host plants in invasive regions also plays an important role in facilitating pest invasions [7]. We tested the hypothesis that rising temperature leads to the enhancement of fitness of invasive species under the same host feeding condition for invasive species.

The fall armyworm (FAW), *Spodoptera frugiperda* (J E Smith) (Lepidoptera: Noctuidae) is one of the most damaging crop pests in the Americas. Its larva can attack more than 353 host plants from 76 families, principally Poaceae (106 taxa) [8]. FAW has long-distance flight capability and is confirmed as an inhabitant in continental Africa (47 countries), Asia (18 countries), and mainland Australia as of February 2020 [9,10]. Due to the absence of diapause and a short generation time, FAW populations can occur all year round in non-native countries with suitable temperatures and host plants [11]. We should explore the fitness of FAW under abiotic (e.g., temperature) and biotic (e.g., host plant) stressors in invasion regions.

FAW is a highly polyphagous herbivore represented by two sympatric host plant strains: a rice strain that feeds on rice and various grasses and a corn strain that primarily feeds on maize, sorghum, cotton, and sugarcane [12,13]. Strain identification analyses based on *COI* and *Tpi* genes showed that a sample of FAW in Yunnan province in China collected in January 2019 belonged to a corn strain [14]. Based on analysis of the *COI* gene, a sample from Chongqing City belonged to the rice strain, while both the rice and corn strains were found in June 2019 when categorized by the *Tpi* gene [15]. This invasive FAW, possibly originating from the offspring of hybrid populations, may pose a risk to different host plants [16]. Understanding the utilization of hybrid populations of FAW to a variety of host plants is necessary to predict the FAW’s annual reproduction and migration [17]. FAW feeds on plants of the Poaceae family, including sorghum, maize, and coix seed [8]. Sorghum is an important dryland food, fodder, and energy crop in China, and the occurrence of FAW on sorghum was observed in June 2019 [18]. Coix seed is a high-quality cereal used both as a food and for medicine in China, where it accounts for more than 70% of the global market. In 2019, FAW was found to damage coix seed in Qichun county, Hubei province, and Xingyi City, Guizhou province in China [19].

The ability of invasive species to regulate their performance is crucial for adapting to new invading environments as the Earth’s climate rapidly changes [20]. Temperature is a central regulating abiotic variable controlling the development, fecundity, and range of invasive species [20]. The developmental rates of invasive insects increase in response to increasing temperatures [21]. Although FAW is not able to overwinter in northern China and North America due to a lack of diapause and suitable host plants, high temperatures trigger FAW outbreaks through an acceleration of their development and reduction in the duration of their life cycle [6,22,23,24]. Therefore, it is essential to evaluate the effects of a temperature gradient on the performance of FAW.

Life history studies of FAW have been undertaken on different host plants (cotton, millet, corn, sorghum, wheat, bermudagrass, soybean, peanut, oilseed rape, sunflower, tomato, pepper, and eggplant) [6,16,17,25,26,27,28,29]. Differences in the suitability of different maize cultivars and varieties of bermudagrass as a host plant for FAW have been recorded [6,25,26]. Climate change alters the responses of FAW to different maize landraces. FAW at high temperatures is more harmful to White Ranchero maize landraces than to Yellow maize landraces [6]. However, the combined role of host plants and temperatures on the entire life history of FAW is poorly understood.

In the current study, we assessed the life history characteristics of FAW on maize (*Zea mays* L. (Poaceae)), sorghum (*Sorghum bicolor* (L.) Moench), and coix seed (Coix lacryma-jobi var. ma-yuen (Rom. Caill.) Stapf (CL)) at three constant temperatures (20, 25, and 30 °C). An age-stage, two-sex life table was used to evaluate and compare FAW responses to different host plants and temperatures.

## 2. Materials and Methods

### 2.1. Insects and Plants

*Spodoptera frugiperda* individuals were originally collected from corn fields on the 22 April 2019 in Guiding County (107.24° N, 26.58° E), Guizhou Province, China. The stock colony was reared for three generations at the Institute of Entomology, Guizhou University on fresh young maize leaves within 15–20 cm in laboratory conditions, at 28 ± 2 °C, 70 ± 5% RH, and a photoperiod of 14:10 (L–D) h. Due to the highly cannibalistic behavior of older larvae, 1st to 3rd instar larvae were reared together in a breeding box (60 × 50 × 40 cm), and 4th to 6th instar larvae were reared individually in breeding boxes (100 mL; depth: 4 cm; bottom diameter: 5.5 cm; top diameter: 6.5 cm). The insects pupated in sterilized soil with 15% water. Cotton balls with 10% honey water were provided for the adult insects as supplemental nutrition. To collect the same age of *S. frugiperda* eggs, 5 pairs of adults were randomly selected from the colonies and placed inside a 1-L cup with Nylon 100 mesh cloth for ventilation and to encourage laying eggs. A cotton ball soaked with a 10% honey solution was placed in the cup to provide food for the adults described by Tian et al. (2020) [30].

The host plants in this study were maize (Jinongnuo No. 7, Beijing Jinnong Technology Co., Ltd., Beijing, China), sorghum (red tassel, Guizhou Hongyingzi Agricultural Technology Development Co., Ltd., Bijie, China) and coix seed (Jinsha Coix). These plants were seeded in a planter tray within potting mix (20 cm depth × 40 cm diameter) and were irrigated twice a week in the greenhouse. Plants with young and healthy leaves (15–20 cm plant height) were used in the experiment.

### 2.2. Life Table Analysis

To evaluate the effect of host plants and temperatures on the life history parameters of *S. frugiperda*, 900 eggs produced by *S. frugiperda* females were maintained in the nylon 100 mesh. Experiments were carried out at 20, 25, and 30 °C on maize, sorghum, and coix seed at 70% ± 5% RH, with a photoperiod of 14:10 (L–D) h within a climate chamber (Jiangnan Instrument Factory, Ningbo, China). One cluster of 100 eggs (less than 24 h old) was prepared on each host plant at each temperature and transferred into a petri dish with wet filter paper (diameter: 9 cm) at the bottom after spraying with sterile distilled water. The petri dish was then placed at a constant temperature, and leaves of a host plant were added. Petri dishes were closed at the top with transparent plastic wrap. An insect pin (No. 5) was used to puncture the plastic wrap 20 times for ventilation. The eggs were checked daily at 8:30 am and 8:30 pm until hatched, and the number of eggs hatched without success or shriveled was considered as death. When FAW eggs hatched (<6 h), they were individually transferred to a plastic box (6 × 5 × 4 cm) with fresh host plant leaves (maize, sorghum, or coix seed) at each temperature. We provided fresh leaves every day for each treatment. When the mature larvae started curling their bodies, indicating they had reached the pre-pupal period, we cleaned away the frass and covered the larvae with a thin layer of soil until eclosion. Newly emerged females and males were paired and placed in the 1-L cups. The plastic cups were furnished with cotton balls with 10% honey water, and the developmental time and mortality rates of immature stages of *S. frugiperda* were recorded twice every day, at 8:30 a.m. and 8:30 p.m. The duration of pre-oviposition, oviposition, fecundity, and longevity of *S. frugiperda* was observed.

The preliminary data for all individuals at different temperatures and on different host plants were organized and analyzed according to an age-stage, two-sex life table [31,32]. For the population parameters, mean values and standard errors of developmental time, mortality, pre-oviposition period, total pre-oviposition period, fecundity, and longevity of *S. frugiperda* were calculated using TWOSEX-MSChart (Chi 2021) [33], estimated with 100,000 bootstrap replicates.

The age-stage survival rate curve (*s_xj_*) shows the probability that a newly oviposited egg will survive to age *x* and stage *j*. The age-specific survival rate (*l_x_*) was approximated as
$$l_{x}=\sum_{j=1}^{m}s_{xj}$$
where *m* is the number of pest stages. Age-specific fecundity (*m_x_*) was represented by the equation
$$m_{x}=\left(\sum_{j=1}^{m}s_{x_{j}}f_{xj}\right)/\sum_{j=1}^{m}s_{x_{j}}$$

The intrinsic rate of increase (*r*) was predicted using the equation
$$r=\sum_{x=0}^{\infty }e^{-r\left(x+1\right)}l_{x}m_{x}$$

The net reproductive rate (*R*_0_) was the total offspring that an individual produced during its lifetime and was depicted as
$$R_{0}=\sum_{x=0}^{\infty }l_{x}m_{x}$$

The gross reproductive rate (*GRR*) was determined as
$$GRR=\sum_{m_{x}}$$

The finite rate of increase (*λ*) was calculated as
$$\lambda =e^{r}$$

The mean generation time (*T*) was defined as the length of time a population requires to increase by *R*_0_-fold (i.e., e*^rt^* = *R*_0_ or λ*^T^* = *R*_0_) at a stable age distribution, and was estimated as
$$T=\left(\ln R_{0}\right)∕r$$

The age-stage life expectancy (*e_xj_*) is the lifespan that an individual of age *x* and stage *j* is expected to live after age *x*. Because individuals of the same age may differ in stage of development, for individuals of the same age, the life expectancy for an individual of age *x* and stage *y* is calculated as
$$e_{\mathrm{xj}}=\sum_{i=x}^{n}\sum_{j=y}^{m}S_{ij}$$
where *n* is the last age of individuals in the population and *m* is the number of stages. $S_{i_{j}}^{\prime }$ is the probability that an individual of age *x* and stage *y* will survive to age *i* and stage *j*, and is calculated by assuming $S_{i_{j}}^{\prime }\;$= 1 [34]. Reproductive value (*v_xj_*) is described as the contribution of an individual of age *x* and age *j* to the future population and was determined as
$$v_{xj}=\left(e^{r\left(x+1\right)}\right)/s_{xj})\sum_{i=x}^{\infty }e^{-r\left(i+1\right)}\sum_{y=j}^{k}s_{iy}^{\prime }f_{iy}$$

For a proper application of the bootstrap technique, the randomization function was used in TWOSEX-MSChart. Moreover, according to Akca et al. (2015) [35], we used 100,000 resamplings to obtain stable estimates of standard errors. A paired bootstrap test was used to detect statistical differences in the mortality at different stages of *S. frugiperda* on different host plants or different temperatures [33].

## 3. Results

### 3.1. Mortality Rate, Developmental Time, Adult Longevity, and Reproduction

The mortality of *S. frugiperda* reared on different host plants at different temperatures is shown in Table 1. Host plants and temperatures significantly influenced the mortality rate of *S. frugiperda* pupae and pre-adults, while no significant differences were found in the mortality of larvae except between temperatures for the 3rd to 5th instar larvae fed sorghum. The mortality rates of *S. frugiperda* pre-adults were significantly influenced by temperatures when fed maize but not on sorghum and coix seed. Host plants induced significant differences in the mortality rate of *S. frugiperda* pre-adults between maize and the other two host plants at 30 °C, while no significant differences were found at 20 °C. A similar trend was also shown in the age-stage specific survival curves (*s_xj_*): the curves for pupa, female, and male fed on maize were higher than those FAW fed on sorghum and coix seed at 30 °C but not at 20 °C (Figure 1).

The developmental duration of *S. frugiperda* reared on different host plants under different temperatures is provided in Table 2. The age-specific survival rate (*l_x_*), fecundity (*m_x_*), and maternity (*l_x_m_x_*) curves of *S. frugiperda* reared on different host plants at different temperatures are shown in Figure 2. In all host plant conditions, the temperature significantly impacted the developmental time of *S. frugiperda* eggs, 1st to 6th instar larvae, pre-pupae, pupae, and pre-adults (*P* < 0.05); when temperature increased, the developmental period of *S. frugiperda* at each stage shortened significantly. At each temperature, the type of host plant did not affect the development time of *S. frugiperda* eggs. There was no significant difference when *S. frugiperda* 1st to 6th instar larvae were fed different host plants at 20 °C, while 2nd to 5th instar larvae developed rapidly on maize at 25 and 30 °C (*P* < 0.05). However, at 20 and 30 °C, the developmental duration of *S. frugiperda* 2nd to 5th instar larvae fed maize was shorter than when fed sorghum and coix seed, while there was no significant difference between sorghum and coix seed. The duration of pre-pupae and pupae were significantly influenced by plant host type at all temperatures, except for the pre-pupae at 25 °C. Feeding on maize significantly shortened the duration of the immature stages of *S. frugiperda* at each temperature.

The adult longevity and reproduction of *S. frugiperda* reared on different host plants at different temperatures are shown in Table 3. Host plants and ambient temperatures significantly influenced the longevity, total pre-oviposition period, and fecundity of FAW. The longevity of FAW females and males was shortest when FAW were fed maize at 30 °C. There were no significant differences in the longevity of FAW when fed sorghum and coix seed at 20 and 25 °C, but a significant difference was observed at 30 °C. Under the same host, the life expectancy (*e_xj_*) of *S. frugiperda* was shortened with increased temperature, showing a significant decreasing trend (Figure 3).

No significant difference was observed in APOP among the three host plants at any temperature. However, the total pre-oviposition period in maize was significantly shorter than that detected under the two other hosts. Female and male adults lived shorter in corn and produced more eggs than those reared on sorghum and coix seed at any temperature. The host plant did not affect the pre-oviposition period (APOP) and oviposition period of FAW at all temperatures. However, a significantly lower total pre-oviposition period (TPOP) was observed on maize host plants. Increasing temperature significantly reduced the APOP and TPOP of FAW, but a shortened oviposition period was observed when FAW was reared on each host plant at 20 °C. Host plants significantly influenced the fecundity of FAW at 25 and 30 °C, but relatively lower temperatures only reduced their fecundity when fed maize. The highest fecundity of FAW was 954.3 and 956.8 when fed maize at 25 and 30 °C, respectively. The highest peaks of *m_x_* that occurred on maize (14.9 offspring), sorghum (13.5 offspring), and coix seed (11.6 offspring) were similar at 20°C, but the highest value of *m_x_* on maize (35.91 offspring) at 30 °C was higher than that on sorghum and coix seed at 20 °C. Similar results were also found in the maternity (*l_x_m_x_*) curves of *S. frugiperda*. Moreover, the relatively lower temperature did not induce a significant difference in the highest reproductive values (*v_xj_*) of pupa and females among maize, sorghum, and coix seed at 20 °C, but feeding on maize at 30 °C had greater reproductive values of pupa and female than those moths fed on sorghum and coix seed (Figure 4).

### 3.2. Life Table Parameters

Host plants and temperatures significantly influenced the intrinsic rate of natural increase (*r*), finite rate of increase (*λ*), and average generation calendar period (*T*) of FAW, except at 20 °C for *r* and *λ* (Table 4). The values of *r*, *λ*, and net reproductive rate (*R*_0_) of FAW significantly increased with increasing temperature on each host plant apart from the *R*_0_ on coix seed. The highest *r*, *λ*, and *R*_0_ of FAW were found on maize at 30 °C. Feeding on maize achieved higher *r*, *λ*, and *R*_0_ at 25 and 30 °C than feeding on sorghum and coix seed. The *T* was significantly shorter when FAW were fed maize at all temperatures, while the gross reproductive rate (*GRR*) was significantly higher than sorghum and coix seed.

## 4. Discussion

Fall armyworm (FAW) is a highly multivoltine polyphagous herbivore that feeds on commercial crops. Higher temperatures reduce its development time, resulting in more generations during the crop growing season [6,26,36,37]. FAW feed on economically important crops, such as rice, sorghum, maize, sugarcane, wheat, millet, cotton, bermudagrass, and different vegetables [6,17,25,26,27,28,29]. Previous studies only considered the independent responses of temperature or host plant on the performance of FAW. There was little information available on the interactive effects of different temperatures and host plants on the entire life cycle of FAW [6,25]. Our study found that both temperature and host plant significantly influenced the survival, development, reproduction, and population growth of FAW. Host plants did not influence the developmental time of larvae, APOP, oviposition period, fecundity, or life table parameters of FAW at relatively low temperature (20 °C). However, significant differences were found at both 25 and 30 °C. Therefore, different responses by FAW were triggered by temperatures to different host plants.

The development rates of *S. frugiperda* increased with increasing temperatures. Optimal temperature can improve the survival rate, accelerate the development rate of immature stages, result in more generations, and enhance the probability of their establishment [2,36,37]. The survival rate of FAW larvae and pupae increases in response to increasing temperature [36,38]. However, different temperatures do not influence the percentage of emerged FAW adults [6]. Our study found a lower mortality of FAW pre-adults under higher temperatures when fed maize. It seems that temperature triggers different survival responses by invasive species depending on the host plant. FAW fed on tomato and potato have significantly lower survival rates at 25 °C compared to maize host plants [39]. In contrast, FAW larva–adult survival does not differ between maize and sorghum [17], cotton, millet, corn, and soybean [27], and between soybean, cotton, maize, wheat, oat leaves, and an artificial diet [40]. Moreover, no differences were found in larva–adult survival rates when fed on corn, sorghum, wheat, soybean, peanut, and cotton [28]. In our study, no significant difference was found in the mortality of FAW pre-adults across the three host plants at 20 °C. However, 30 °C caused a significantly lower mortality rate of FAW on maize than on sorghum and coix seed, which is consistent with du Plessi’s results [36]. The differential response of host plants to increasing temperature may cause different mortality rates of FAW. Therefore, it was critical to evaluate the performance of invasive species on different host plants at different temperatures.

The developmental time of the immature stages of FAW is significantly influenced by temperature and the type of host plant [6,25,26,36,38,40]. At higher temperatures, the development period of FAW is significantly reduced [26,36,38]. In the present study, we found that increasing temperatures significantly shortened the developmental time of FAW eggs, larvae, pre-pupae, pupae, and pre-adults on each host plant, which significantly accelerated generations. However, temperature caused differences in FAW developmental responses to host plants. There were no significant differences in the developmental time of 2nd to 6th instar larvae at 20 °C, but there was a significant reduction in developmental time on maize host plant at 30 °C. The duration of the larval stage of FAW fed four varieties of bermudagrass at 20 and 30 °C shows the opposite tendency [25] These differences in FAW development based on temperature and host plant type might help predict FAW occurrence in both normal and extreme climatic conditions.

Temperature and the type of host plant significantly influenced the reproduction parameters of FAW [25,26,28,29,38,40]. Increased temperature (30 °C) significantly reduced the longevity of FAW fed on corn, while there was no longevity reduction when the temperature was increased from 21 to 25 °C [26]. Our study found temperature triggered different responses of longevity, TPOP, and fecundity of FAW fed on host plants. No significant differences in longevity and TPOP were found between sorghum and coix seed at 20 and 25 °C, but at 30 °C, the longevity was significantly reduced in FAW fed on coix seed. Higher temperatures increased FAW fecundity when fed on maize. An ambient temperature of 30 °C significantly increased the fecundity of FAW across the three host plants, but there was no significant difference at 20 °C.

Assessment of population parameters is an essential method for evaluating the effect of temperatures and host plants on the fitness of invasive insects [41]. No previous studies have recorded the life table parameters of FAW under different temperatures [6,38]. Our study first reported that increasing temperature significantly affected the population parameters (*R*_0_, *r*, *λ*, and *GRR*) of FAW on each host plant, except for the *R*_0_ when fed coix seed and *GRR* when fed sorghum host plants. Host plants significantly influenced the population parameters for FAW [16,27,29,42]. The values of *r* and *λ* for FAW fed on maize are significantly higher than those for wheat, soybean, Chinese cabbage, tomato, and pepper [29,42]. FAW fed on millet have higher *R*_0_ and *r* than on cotton and soybean plants [27]. Our study found similar results that feeding on maize caused higher *r*, *λ*, and *R*_0_ for FAW at 25 and 30 °C than feeding on sorghum and coix seed, but at 20 °C, this trend did not reach statistical significance. Therefore, temperature also influenced the population parameter responses of FAW to the host plants.

Climate anomalies are important drivers for expanding habitat distribution and elevating the invasion risk of invasive species [43]. For example, temperature-driven changes caused insect outbreaks by affecting their life history traits to alter system stability [4]. FAW is a poikilothermic pest species that could regulate growth and development with a temperature change. In our study, the increased temperature caused changes in the life history of FAW in each host plant. However, relatively low temperatures (20 °C) appeared to eliminate temperature-mediated synergism in FAW performance. There were significant life stage effects at 20 °C [44,45]. Feeding on different host species also influenced the cold-tolerance strategy of *Epiphyas postvittana* (Walker) [46]. Furthermore, FAW collected in China may be the progeny of hybrid populations, which should have developed substantial adaptations to maize plants [16]. Currently, there is little information available on FAW adaptation to coix seeds throughout the world [19]. Adaptation to local host plants may influence the response of the oak specialist butterfly (*Erynnis propertius*) to climate change [47]. Therefore, strong adaptation to a local host plant (maize) might increase responses to increasing temperature. However, this research still had many limitations. We only determined the performance of FAW on different host plants at 20 °C, 25 °C, and 30 °C under the laboratory condition, and further study should focus on temperatures below 20 °C or above 30 °C. Meanwhile, fluctuations and constant temperatures have different effects on the development and longevity of insects. The insects reared at fluctuating profiles with lower average temperatures develop faster and survive longer than those reared at constant temperatures, while higher average fluctuating temperatures result in longer developmental duration and lower longevity of insects [48,49]. In our study, we only tested the constant temperatures on FAW without considering fluctuating temperatures in the laboratory. We hope that the result stimulates future study on this topic to evaluate the fluctuating temperatures on the occurrence of FAW on different host plants in the field.

## 5. Conclusions

In conclusion, our experiment carried out detailed observation of each stage after 12 h, and the results may be more accurate than those observed once in 24 h. Meanwhile, our study revealed that temperature and the type of host plant significantly influenced the performance of FAW, and the fitness of FAW on different host plants varied depending on temperature. Lower temperatures could eliminate temperature-mediated synergism of FAW on local host plants and speed up their adaptation to plants in new locales, which may also result in outbreaks of invasive species. In our study, we conducted detached leaf assays to detect the life history of FAW under different temperatures in laboratory conditions. Further studies should focus on the interactions between temperatures and host plant species outside of the lab; we recommend the next step being the greenhouse.

## Figures and Tables

**Figure 1 insects-13-00211-f001:**
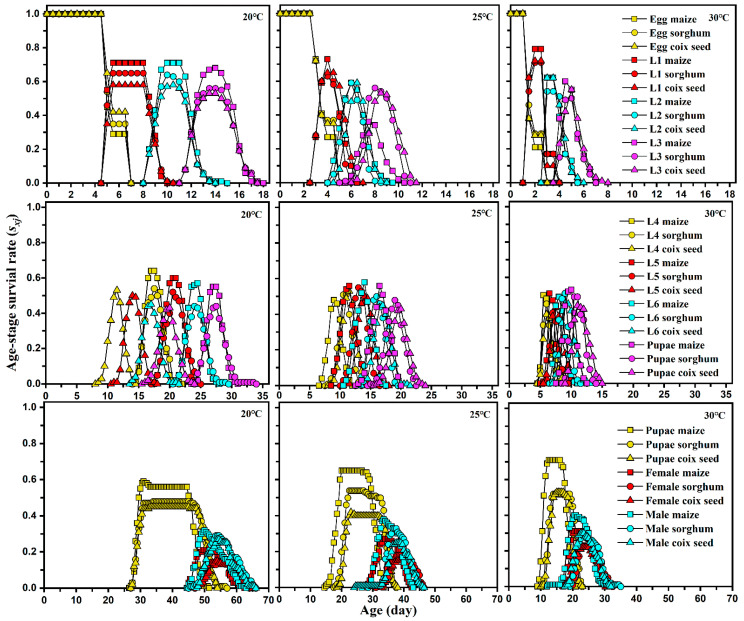
Age-stage specific survival rate (*s_xj_*) of each developmental stage of *Spodoptera frugiperda* feeding on different host plants at different temperatures.

**Figure 2 insects-13-00211-f002:**
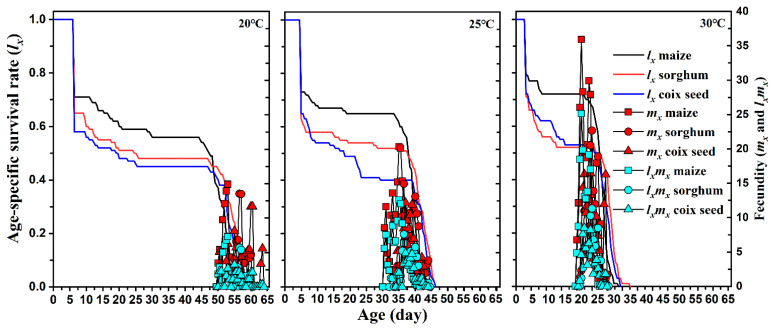
Age-stage specific survival rate (*l_x_*), fecundity (*m_x_*), and maternity (*l_x_m_x_*) of *Spodoptera frugiperda* on different hosts and at different temperatures.

**Figure 3 insects-13-00211-f003:**
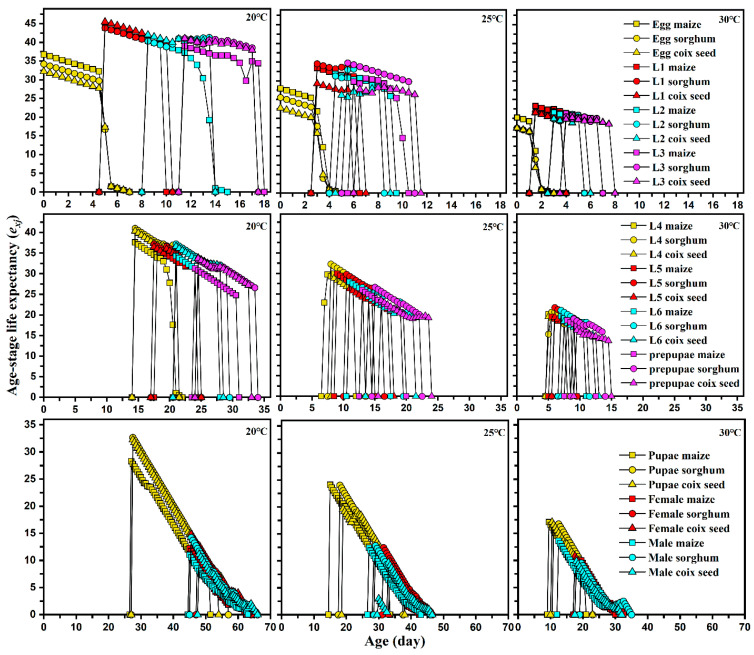
Age-stage specific life expectancy (*e_xj_*) of each development stage of *Spodoptera frugiperda* reared on different hosts at different temperatures.

**Figure 4 insects-13-00211-f004:**
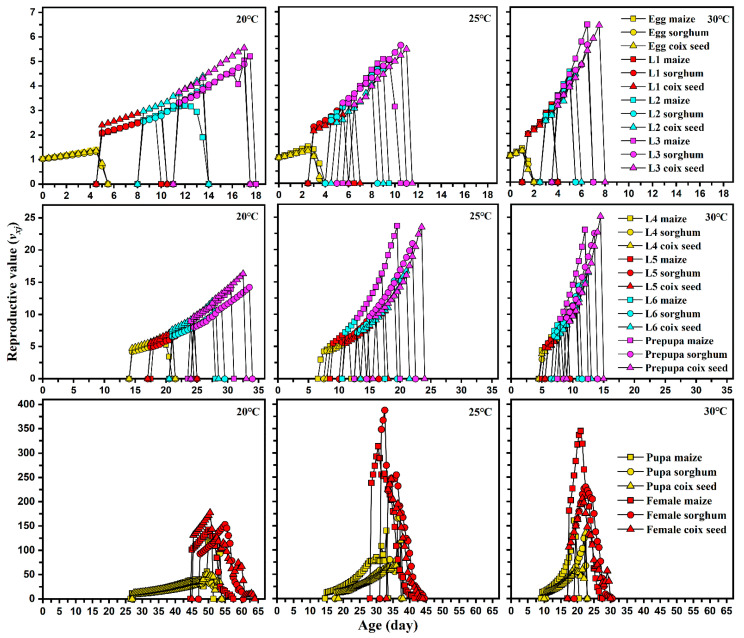
Age-stage reproductive value (*v_xj_*) of each developmental stage of *Spodoptera frugiperda* on different hosts and at different temperatures.

**Table 1 insects-13-00211-t001:** Mortality of mean (±SE) different developmental stages of *Spodoptera frugiperda* reared on different host plants at different *temperatures*.

	Mortality Rate					
Temperature (°C)	Egg			1st instar Larva		
	Maize	Sorghum	Coix seed	Maize	Sorghum	Coix seed
20	0.29 ± 0.045 aA	0.35 ± 0.048 aA	0.42 ± 0.049 aA	0 ± 0 aA	0 ± 0 aA	0 ± 0 aA
25	0.27 ± 0.044 aA	0.37 ± 0.05 aA	0.35 ± 0.048 aAB	0 ± 0 aA	0 ± 0 aA	0 ± 0 aA
30	0.21 ± 0.041 aA	0.29 ± 0.046 aA	0.28 ± 0.045 aB	0 ± 0 aA	0 ± 0 aA	0 ± 0 aA
Temperature (°C)	2nd instar larva			3rd instar larva		
	Maize	Sorghum	Coix seed	Maize	Sorghum	Coix seed
20	0.03 ± 0.017 aA	0.06 ± 0.024 aA	0.03 ± 0.017 aA	0.03 ± 0.017 aA	0.04 ± 0.02 aA	0.03 ± 0.017 aA
25	0.02 ± 0.014 aA	0.05 ± 0.022 aA	0.07 ± 0.026 aA	0.01 ± 0.009 abA	0 ± 0 bB	0.04 ± 0.02 aA
30	0.02 ± 0.014 aA	0.05 ± 0.022 aA	0.05 ± 0.022 aA	0 ± 0 aA	0.03 ± 0.017 aAB	0.02 ± 0.014 aA
Temperature (°C)	4th instar larva			5th instar larva		
	Maize	Sorghum	Coix seed	Maize	Sorghum	Coix seed
20	0.04 ± 0.02 bA	0 ± 0 aB	0 ± 0 aA	0.02 ± 0.014 aAB	0.04 ± 0.02 aA	0.04 ± 0.02 aA
25	0.03 ± 0.017 aAB	0 ± 0 aB	0 ± 0 aA	0 ± 0 aB	0 ± 0 aB	0.02 ± 0.014 aA
30	0 ± 0 bB	0.06 ± 0.024 aA	0.02 ± 0.014 bA	0.05 ± 0.022 aA	0.01 ± 0.01 bAB	0.01 ± 0.01 bA
Temperature (°C)	6th instar larva			Pre-pupa		
	Maize	Sorghum	Coix seed	Maize	Sorghum	Coix seed
20	0 ± 0 aA	0 ± 0 aA	0.01 ± 0.01 aA	0 ± 0 aA	0.03 ± 0.017 aA	0.02 ± 0.014 aB
25	0 ± 0 bA	0.03 ± 0.017 aA	0.03 ± 0.017 aA	0.01 ± 0.01 aA	0.01 ± 0.01 aA	0 ± 0 aB
30	0 ± 0 aA	0 ± 0 aA	0 ± 0 aA	0 ± 0 bA	0.02 ± 0.014 bA	0.07 ± 0.026 aA
Temperature (°C)	Pupa			Pre-adult		
	Maize	Sorghum	Coix seed	Maize	Sorghum	Coix seed
20	0.03 ± 0.017 aA	0 ± 0 aA	0 ± 0 aB	0.44 ± 0.05 aA	0.52 ± 0.05 aA	0.55 ± 0.05 aA
25	0.01 ± 0.01 bA	0.02 ± 0.014 bA	0.08 ± 0.027 aA	0.35 ± 0.048 bAB	0.48 ± 0.05 bA	0.59 ± 0.05 aA
30	0 ± 0 aA	0.02 ± 0.014 aA	0.02 ± 0.014 aB	0.28 ± 0.045 bB	0.48 ± 0.05 aA	0.47 ± 0.05 aA

The lowercase letters (a–c) indicate significant differences in mortality at different developmental stages of *Spodoptera frugiperda* on different hosts under the same temperature. The capital letters (A–C) indicate significant differences in mortality at different developmental stages of *S**. frugiperda* at different temperatures with the same host plant. A paired bootstrap test was used to detect statistical differences in the mortality at different stages of *S. frugiperda* on different host plants or different temperatures. Standard errors were estimated from 100,000 bootstrap resampling.

**Table 2 insects-13-00211-t002:** Duration of the mean (±SE) each developmental stage of *Spodoptera frugiperda* reared on different host plants under different temperatures.

	Developmental Time (d)											
Temperature (°C)	Egg							1st instar larva				
	N	Maize	N	Sorghum	N	Coix seed	N	Maize	N	Sorghum	N	Coix seed
20	71	5.11 ± 0.02 aA	65	5.15 ± 0.03 aA	58	5.2 ± 0.03 aA	71	3.97 ± 0.05 aA	65	3.96 ± 0.06 aA	58	3.92 ± 0.06 aA
25	73	3.41 ± 0.04 aB	63	3.30 ± 0.04 aB	65	3.32 ± 0.04 aB	73	2.01 ± 0.05 bB	63	2.06 ± 0.04 bB	65	2.53 ± 0.05 aB
30	79	1.61 ± 0.02 aC	71	1.62 ± 0.03 aC	72	1.57 ± 0.02 aC	79	1.61 ± 0.02 aC	71	1.62 ± 0.03 aC	72	1.57 ± 0.02 aC
Temperature (°C)	2nd instar larva						3rd instar larva					
	N	Maize	N	Sorghum	N	Coix seed	N	Maize	N	Sorghum	N	Coix seed
20	68	3.29 ± 0.04 aA	59	3.21 ± 0.03 aA	55	3.21 ± 0.03 aA	65	3.58 ± 0.05 aA	55	3.7 ± 0.06 aA	52	3.71 ± 0.06 aA
25	71	1.98 ± 0.04 aB	58	1.91 ± 0.05abB	58	2.01 ± 0.03 bB	70	1.14 ± 0.03 bB	58	2.63 ± 0.05 aB	54	2.64 ± 0.05 aB
30	77	1.23 ± 0.03 cC	66	1.33 ± 0.03 bC	67	1.46 ± 0.05 aC	77	1.32 ± 0.03 bC	63	1.24 ± 0.03 bC	65	1.45 ± 0.04 aC
Temperature (°C)		4th instar larva					5th instar larva					
	N	Maize	N	Sorghum	N	Coix seed	N	Maize	N	Sorghum	N	Coix seed
20	61	3.32 ± 0.04 aA	55	3.38 ± 0.05 aA	52	3.38 ± 0.05 aA	59	3.54 ± 0.04 aA	51	3.54 ± 0.06 aA	48	3.46 ± 0.06 aA
25	67	1.84 ± 0.03 cB	58	2.17 ± 0.03 bB	54	2.58 ± 0.04 aB	67	2.43 ± 0.06 bB	58	2.78 ± 0.06 aB	52	2.88 ± 0.06 aB
30	77	0.96 ± 0.02 bC	57	1.18 ± 0.03 aC	63	1.18 ± 0.03 aC	72	1.08 ± 0.02 bC	56	1.12 ± 0.03 bC	62	1.59 ± 0.03 aC
Temperature (°C)		6th instar larva					Pre-pupa					
	N	Maize	N	Sorghum	N	Coix seed	N	Maize	N	Sorghum	N	Coix seed
20	59	3.18 ± 0.03 aA	51	3.14 ± 0.05 aA	47	3.2 ± 0.05 aA	59	3.01 ± 0.04 bA	48	3.17 ± 0.05 aA	45	3.2 ± 0.05 aA
25	67	2.67 ± 0.05 aB	55	2.81 ± 0.07 aB	49	2.8 ± 0.07 aB	66	2.8 ± 0.06 aB	54	2.96 ± 0.06 aB	49	2.89 ± 0.07 aB
30	72	1.34 ± 0.05 bC	56	1.93 ± 0.04 aC	62	1.93 ± 0.02 aC	72	1.76 ± 0.04 bC	54	2.36 ± 0.07 aC	55	2.27 ± 0.07 aC
Temperature (°C)		Pupa					Pre-adult					
	N	Maize	N	Sorghum	N	Coix seed	N	Maize	N	Sorghum	N	Coix seed
20	56	18.36 ± 0.17 bA	48	21.47 ± 0.24 aA	45	21.53 ± 0.23 aA	56	47.34 ± 0.19 bA	48	50.75 ± 0.3 aA	45	50.82 ± 0.28 aA
25	65	12.83 ± 0.08 bB	52	13.92 ± 0.13 aB	41	13.56 ± 0.33 aB	65	31.09 ± 0.17 bB	52	34.59 ± 0.2 aB	41	35.02 ± 0.32 aB
30	72	7.69 ± 0.11 bC	52	8.87 ± 0.09 aC	53	7.48 ± 0.09 bC	72	18.59 ± 0.15 cC	52	21.3 ± 0.13 aC	53	20.42 ± 0.17 bC

The lowercase letters (a–c) indicate significant differences in the developmental duration of *Spodoptera frugiperda* on different hosts at the same temperature. The capital letters (A–C) indicate significant differences in the developmental duration of *S. frugiperda* at different temperatures on the same host. A paired bootstrap test was used to detect statistical differences in the mortality at different stages of *S. frugiperda* on different host plants or different temperatures. Standard errors were estimated using 100,000 bootstrap resampling.

**Table 3 insects-13-00211-t003:** Adult longevity and reproductive parameters mean (±SE) of *Spodoptera frugiperda* reared on different host plants under different temperatures.

Temperature (°C)	Adult Longevity/d (Female)			Adult Longevity/d (Male)		
	Maize	Sorghum	Coix seed	Maize	Sorghum	Coix seed
20	57.2 ± 0.54 bA	60.53 ± 0.64 aA	60.34 ± 0.94 aA	56.1 ± 0.63 bA	59.95 ± 0.65 aA	59.4 ± 0.62 aA
25	39.87 ± 0.33 bB	43.94 ± 0.5 aB	43.4 ± 0.46 aB	39.41 ± 0.37 bB	41.87 ± 0.41 aB	42.19 ± 0.62 aB
30	27.18 ± 0.28 cC	29.48 ± 0.33 aC	28.34 ± 0.4 bC	26.1 ± 0.32 cC	29.05 ± 0.41 aC	27.89 ± 0.41 bC
Temperature (°C)	Total pre-oviposition period (TPOP)/d			Adult pre-oviposition period (APOP)/d		
	Maize	Sorghum	Coix seed	Maize	Sorghum	Coix seed
20	51.72 ± 0.33 bA	56 ± 0.26 aA	55.14 ± 1.14 aA	5.22 ± 0.15 aA	5.5 ± 0.34 aA	5.14 ± 0.14 aA
25	32.62 ± 0.5 bB	36.35 ± 0.65 aB	36.67 ± 0.49 aB	1.83 ± 0.21 aB	2.3 ± 0.4 aB	1.67 ± 0.21 aB
30	21.14 ± 0.17 cC	23.86 ± 0.39 aC	22.67 ± 0.42 bC	2.17 ± 0.19 aB	2.43 ± 0.25 aB	2.42 ± 0.26 aB
Temperature (°C)	Oviposition days/*O_d_*			Fecundity (no. of eggs)		
	Maize	Sorghum	Coix seed	Maize	Sorghum	Coix seed
20	1.78 ± 0.15 aC	1.83 ± 0.17 aB	1.86 ± 0.14 aB	404.56.95 ± 49.16 aB	426.83 ± 76.68 aA	415.71 ± 73.41 aB
25	3.92 ± 0.29 aA	3.4 ± 0.34 aA	3.33 ± 0.21aA	954.92 ± 52.83 aA	626.87 ± 95.21 bA	628.17 ± 52.01 bA
30	2.61 ± 0.31 aB	2.71 ± 0.27 aA	2.83 ± 0.27 aA	956.83 ± 93.5 aA	615 ± 77.97 bA	558.42 ± 66.22 bAB

The lowercase letters (a–c) indicate significant differences in the developmental duration of *Spodoptera frugiperda* on different hosts at the same temperature. The capital letters (A–C) indicate significant differences in the developmental duration of *S. frugiperda* at different temperatures on the same host. A paired bootstrap test was used to detect statistical differences in the mortality at different stages of *S. frugiperda* on different host plants or different temperatures. Standard errors were estimated using 100,000 bootstrap resampling.

**Table 4 insects-13-00211-t004:** The intrinsic rate of increase (*r*), finite rate of increase (*λ*), net reproductive rate (*R*_0_), mean generation time (*T*), and gross reproductive rate (*GRR*) (±SE) of *Spodoptera frugiperda* reared on different host plants at different temperatures.

Temperature (°C)	*r* (d^−1^)					*λ* (d^−1^)
	Maize	Sorghum	Coix seed	Maize	Sorghum	Coix seed
20	0.068 ± 0.007 aC	0.057 ± 0.009 aC	0.06 ± 0.009 aC	1.07 ± 0.008 aC	1.058 ± 0.009 aC	1.062 ± 0.009 aC
25	0.137 ± 0.009 aB	0.108 ± 0.009 bB	0.094 ± 0.012 bB	1.146 ± 0.01 aB	1.114 ± 0.01 bB	1.099 ± 0.013 bB
30	0.224 ± 0.011 aA	0.176 ± 0.012 bA	0.173 ± 0.013 bA	1.251 ± 0.013 aA	1.192 ± 0.014 bA	1.189 ± 0.016 bA
Temperature (°C)	*R*_0_ (offspring)			*T* (d)		
	Maize	Sorghum	Coix seed	Maize	Sorghum	Coix seed
20	36.41 ± 12.309 aB	25.61 ± 10.913 aB	29.1 ± 11.64 aA	52.843 ± 0.383 bA	57.38 ± 0.316 aA	55.743 ± 1.47 aA
25	114.59 ± 31.535 aA	62.687 ± 20.821 abAB	37.69 ± 15.065 bA	34.7 ± 0.496 bB	38.39 ± 0.599 aB	38.48 ± 0.494 aB
30	172.23 ± 40.121 aA	86.1 ± 23.755 abA	67.01 ± 19.699 bA	23.017 ± 0.176 bC	25.338 ± 0.294 aC	24.353 ± 0.429 aB
Temperature (°C)	*GRR* (offspring)					
	Maize	Sorghum	Coix seed			
20	76.67 ± 25.008 aB	67.04 ± 27.34 aB	96.65 ± 38.96 aA			
25	193.03 ± 51.027 aA	141.88 ± 46.353 aAB	100.01 ± 38.056 aA			
30	265.07 ± 59.009 aA	179.88 ± 45.955 aA	161.22 ± 47.385 aA			

The lowercase letters (a–c) indicate significant differences in the developmental duration of *Spodoptera frugiperda* on different hosts at the same temperature. The capital letters (A–C) indicate significant differences in the developmental duration of *S. frugiperda* at different temperatures on the same host. A paired bootstrap test was used to detect statistical differences in the mortality at different stages of *S. frugiperda* on different host plants or different temperatures. Standard errors were estimated using 100,000 bootstrap resampling.

## Data Availability

Dataset is available from the first author on reasonable request.

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
