# Peer review of "The Effect of Temperatures and Hosts on the Life Cycle of Spodoptera frugiperda (Lepidoptera: Noctuidae)"

_insects, 2022, doi:10.3390/insects13020211_

Round 1
Reviewer 1 Report
Manuscript: Temperature triggers different responses in invasive species to host plant by Chen et al. (Insects).
Overview
This paper provides evidence on Spodoptera frugiperda's performance on three hosts under three constant temperatures. The authors found that species performance on the same host varies according to temperature, with the high temperature triggering different responses among hosts, while it was not observed under low ones. Thus, these findings provide evidence on how fitness cost is influenced by environmental stress (high temperatures) and host-plant quality.
Yet, the paper presents several arguments that are not supported by the findings as to the author's use of climate change as the main idea, which in my view, was not tested here. For example, in lines 49-51, the author's arguments ‘We tested the hypothesis that rising temperature combined with a lack of co-evolved plants leads enhanced fitness for invasive species, promoting pest invasions.` However, the rising temperature refers to three temperatures tested (20, 25, and 30 °C) that are a common approach for insect development time experiments (e.g., Prasad et al. 2021). Also, no justification is provided about why the authors chose this range of temperature since they used it to conclude that different responses of FAW were triggered by climate change (lines 297-298). In addition, the host presence is related to pest establishment, not invasions.
I would recommend that the authors revise these main arguments and adjust the discussion of climate change in the paper as there is no evidence. I have outlined other comments below and hope they can be helpful to the authors. Thus, I recommend this paper for publication after a major review.
Prasad, T., Srinivasa Rao, M., Rao, K., Bal, S., Muttapa, Y., Choudhary, J., & Singh, V. (2021). Temperature-based phenology model for predicting the present and future establishment and distribution of recently invasive Spodoptera frugiperda (J. E. Smith) in India. Bulletin of Entomological Research, 1-15. doi:10.1017/S0007485321000882
Comments
L. 15-20: The summary is fragmented, and sentences appear not connected with speculative conclusions that were not tested here. For instance, in lines
L. 17-18 the authors said that low temperatures could induce the fitness of FAW when colonizing new host plants. But the low temperature refers to 20°C. Is it a low temperature? Potentially suitable areas for S. frugiperda in China, for example, include a range of temperature from 13 to 36 °C (Wang et al. 2020). Which new host plant do the authors refer to? Does the new host was tested here?
Wang et al., 2020. Potential distribution of Spodoptera frugiperda (J.E. Smith) in China and the major factors influencing distribution. Global Ecology and Conservation, DOI: https://doi.org/10.1016/j.gecco.2019.e00865
L. 58-61: Sentence appears fragmented. Please revise.
L. 93-94: It is not valid. The authors would like to say that it is not performed at the same study, as in discussion they mention other studies to support the results found, as seen in lines 299-312 and 323-325.
Figures 1-4: The figures are of low quality. We cannot read the legend of the colors provided. Would you please give a better-quality one?
L. 314-315: Please provide a reference to ‘different temperatures representing extreme climate change in recent years’. Do the authors believe that 30 °C is an extreme scenario?
Author Response
Response to Reviewer 1 Comments
Point 1: lines 49-51, Yet, the paper presents several arguments that are not supported by the findings as to the author's use of climate change as the main idea, which in my view, was not tested here. For example, the author's arguments ‘We tested the hypothesis that rising temperature combined with a lack of co-evolved plants leads enhanced fitness for invasive species, promoting pest invasions.’ However, the rising temperature refers to three temperatures tested (20, 25, and 30 °C) that are a common approach for insect development time experiments (e.g., Prasad et al. 2021).
Response 1: Yes, we agree with your question. We changed to “We tested the hypothesis that rising temperature leads to the enhancement of fitness of invasive species under the same host feeding condition for invasive species”
Point 2: lines 297-298, Also, no justification is provided about why the authors chose this range of temperature since they used it to conclude that different responses of FAW were triggered by climate change. In addition, the host presence is related to pest establishment, not invasions.
Response 2: Yes, we agreed with your comments. We had deleted the description of climate change in the text. As for the range of temperatures used in this ms, we based on the average temperature from April to August (ranged from 20 to 30°C) in the planting area of maize, sorghum, and coix seed in Xingyi City and Bijie City, Guizhou Province, China.
Point 3: L. 15-20: The summary is fragmented, and sentences appear not connected with speculative conclusions that were not tested here. For instance, in lines
Response 3: Yes, we agreed with your comment. We changed to “FAW is a worldwide agricultural pest that seriously threatens the safety of grain production. In this study, both temperature and host plant significantly influenced the survival, development, reproduction, and population growth of FAW. Host plants did not significantly influence the developmental time of larvae, APOP, oviposition period, fecundity, or life table parameters of FAW at temperature (20°C). However, increasing temperature and host plant significantly affected the performance of FAW to varying degrees, significant differences were found at both 25℃ and 30°C. At relatively low temperatures (20°C) could induce the fitness of FAW when colonizing new host plants in invasion regions.”
Point 4: L. 17-18 the authors said that low temperatures could induce the fitness of FAW when colonizing new host plants. But the low temperature refers to 20°C. Is it a low temperature? Potentially suitable areas for S. frugiperda in China, for example, include a range of temperature from 13 to 36 °C (Wang et al. 2020). Which new host plant do the authors refer to? Does the new host was tested here?
Response 4: we agreed with your comment. We used three temperatures (20, 25, and 30°C) in this ms based on the average temperatures of host plants (corn, sorghum, and coix seed) from April to August in Xingyi City and Bijie City, Guizhou Province, China. We changed “low temperature” to “relatively low temperature” in the text. We just tried to show low temperature (20°C) compared with 30°C. We had revised the simple summary again and provided the host plants (Corn, sorghum, and coix seed) in the paragraph.
Point 5: L. 58-61: Sentence appears fragmented. Please revise.
Response 5: We agreed with your comment. We deleted one sentence and revised another sentence. “We should explore the fitness of FAW under abiotic (e.g., temperature) and biotic (e.g., host plant) stressors in invasion regions.”
Point 6: L. 93-94: It is not valid. The authors would like to say that it is not performed at the same study, as in discussion they mention other studies to support the results found, as seen in lines 299-312 and 323-325.
Response 6: We agreed with your comment. We had changed to “However, the combined role of host plants and temperatures on the entire life history of FAW is poorly understood.” In the introduction. Yes, some articles had studied the effect of temperature or host plants on the development of FAW. But this ms was the first research on the combined effect of temperature and host plants on the entire life history of FAW.
Point 7: Figures 1-4: The figures are of low quality. We cannot read the legend of the colors provided. Would you please give a better-quality one?
Response 7: Yes, many thanks for we improved the quality of the figures.
Figure 1. Age-stage specific survival rate (sxj) of each developmental stage of Spodoptera frugiperda feeding on different host plants at different temperatures.
Figure 2. Age-stage specific survival rate (lx), fecundity (mx), and maternity (lxmx) of Spodoptera frugiperda on different hosts and at different temperatures.
Figure 3. Age-stage specific life expectancy (exj) of each development stage of Spodoptera frugiperda reared on different hosts at different temperatures.
Figure 4. Age-stage reproductive value (vxj) of each developmental stage of Spodoptera frugiperda on different hosts and at different temperatures.
Point 8: L. 314-315: Please provide a reference to ‘different temperatures representing extreme climate change in recent years’. Do the authors believe that 30 °C is an extreme scenario?
Response 8: We agreed with your comment. We had deleted this part and changed to “Therefore, it was critical to evaluate the performance of invasive species on different host plants under different temperatures.”

Reviewer 2 Report
This article by Chen and colleagues (insects-1564720) is well motivated, the structure is appropriate, and the manuscript is well written without missing any key details. The methods used are appropriate for the objectives of the work and, in general, well depicted. The resulting figures are sufficient, informative, and of good quality helping to follow the reasoning throughout the manuscript. The discussion of results and comments on future research could be improved in my opinion.
A few remarks have been made below for authors to consider.
Firstly, the word template should be changed from Biology MDPI to Insects MDPI.
My main concern is that the authors are extrapolating the applicability of their results beyond what the design supports. The datasets presented here are derived from three sets of highly artificial constant temperatures of 20C, 25C, and 30C most of which are within the optimal range for development and survival of FAW (favored temperatures for development, longevity, and survival of FAW are 25-30C, see studies by du Plessis et al. 2020 doi:10.3390/insects11040228; and Garcia et al. https://doi.org/10.1093/jee/tox329). The inference power of the paper is very limited, but the authors do not acknowledge this detail at all and need to be more forthcoming. The egg-to-adult development rate and survival of FAW at temperatures lower than 20C and higher than 30C, which are more critical for the survival of this pest were not investigated. This is a critical limitation of the study, and the authors must concede and discuss this.
Studies across a broader set of constant and fluctuating temperatures are still necessary to understand the real effect of temperature on the characteristics of FAW, as this is the closest to the daily temperature fluctuations that occur in the field. Some of the authors statements would also be much stronger if they tie their work to the body of literature that has built up from rearing insects at temperatures that fluctuate over 24h cycles (see Milosavljevic et al. https://doi.org/10.1093/jee/toy429; and McCalla et al. https://doi.org/10.1093/jee/toz067). The interaction of cyclic temperatures with nonlinear development or life history parameters can introduce significant deviations from results obtained from rearing insects at constant temperatures, especially at the lower and higher temperatures of the development rate, viability, and reproductive activity functions. The consequences of getting this wrong will affect real people and livelihoods. So, I am suggesting to the authors to tone-down the language a little and admit that there are still substantive uncertainties to be considered, including uncertainty as to how generalizable the results are to open field conditions. This is not to diminish the data gathered in this study, they are of value. But it is important for the authors not to overgeneralize, and to warn the reader, including regulatory agencies, against doing so as well. Adding these details will improve the paper in my opinion.
Good luck!
Author Response
Response to Reviewer 2 Comments
Point 1: Firstly, the word template should be changed from Biology MDPI to Insects MDPI.
Response 1: Yes, we agreed with your advice, and changed to Insects.
Point 2: My main concern is that the authors are extrapolating the applicability of their results beyond what the design supports. The datasets presented here are derived from three sets of highly artificial constant temperatures of 20C, 25C, and 30C most of which are within the optimal range for development and survival of FAW (favored temperatures for development, longevity, and survival of FAW are 25-30C, see studies by du Plessis et al. 2020 doi:10.3390/insects11040228; and Garcia et al. https://doi.org/10.1093/jee/tox329). The inference power of the paper is very limited, but the authors do not acknowledge this detail at all and need to be more forthcoming.
Response 2: Yes, many thanks for your question. Line 303-319: we had cited above two papers and changed to “The development rate of S. frugiperda increased linearly with increasing temperatures, especially in the optimal range can improve the survival rate, accelerate egg, larval, egg- adult developmental rate, and reduce mortality of invasive species, resulting in more generations and enhancing the probability of their establishment [2,37-38,40]” some discussion to support our results.
Point 3: The egg-to-adult development rate and survival of FAW at temperatures lower than 20C and higher than 30C, which are more critical for the survival of this pest were not investigated. This is a critical limitation of the study, and the authors must concede and discuss this.
Response 3: Yes, many thanks for your question. Line 372-376, we admit the deficiency, and changed to “However, this study still has many limitations, we only studied the development rate and survival rate of different hosts at 20°C ,25°C and 30 °C, which is an important limitation of the study. We should further explore the development rate and survival rate below 20 °C and above 30 °C, and could objectively and effectively discuss which temperature is more suitable for the development and survival of S. frugiperda.”
Point 4: Studies across a broader set of constant and fluctuating temperatures are still necessary to understand the real effect of temperature on the characteristics of FAW, as this is the closest to the daily temperature fluctuations that occur in the field.
Response 4: Yes, many thanks you for your advice. In this experiment, we try to understand the combined effect of constant temperatures and host plants on the performance of FAW. In the future research, we will focus on the effect of constant and fluctuating temperatures on FAW.
Point 5: Some of the authors statements would also be much stronger if they tie their work to the body of literature that has built up from rearing insects at temperatures that fluctuate over 24h cycles (see Milosavljevic et al. https://doi.org/10.1093/jee/toy429; and McCalla et al. https://doi.org/10.1093/jee/toz067). The interaction of cyclic temperatures with nonlinear development or life history parameters can introduce significant deviations from results obtained from rearing insects at constant temperatures, especially at the lower and higher temperatures of the development rate, viability, and reproductive activity functions. The consequences of getting this wrong will affect real people and livelihoods. So, I am suggesting to the authors to tone-down the language a little and admit that there are still substantive uncertainties to be considered, including uncertainty as to how generalizable the results are to open field conditions. This is not to diminish the data gathered in this study, they are of value. But it is important for the authors not to overgeneralize, and to warn the reader, including regulatory agencies, against doing so as well. Adding these details will improve the paper in my opinion.
Response 5: Yes, many thanks for your question. Line 377-383, we admit that there are still many deficiencies, and changed to “Meanwhile, fluctuations and constant temperatures have different effects on the development and longevity of insects, which reared at fluctuating profiles with lower average temperatures develop faster and survive longer than at constant temperatures. In contrast, higher average fluctuating temperatures can produce insects with longer developmental duration and lower longevity [50,51]. In our article, we only studied the constant temperature without studying the influence of fluctuating temperatures on FAW. Future research may study the influence of fluctuating temperatures on the life history of FAW. In addition, our article combines the effects of different temperatures and three different host plants crosses on the whole life cycle of S. frugiperda, which is different from previous studies. However, this paper is only conducted in the laboratory, and lacks data on the impact of outdoor temperatures and hosts on the life history of S. frugiperda. Therefore, our conclusion is only based on indoor conditions to pro-vide some support for the theory of the whole life cycle of S. frugiperda.”

Round 2
Reviewer 2 Report
The authors have done a fine job addressing my original comments, thank you.
Author Response
Dear Reviewer
Many thanks for addressing comments and suggestions in my manuscript!
Kind regards,
Yichai Chen